# Virtual Screening and Binding Analysis of Potential CD58 Inhibitors in Colorectal Cancer (CRC)

**DOI:** 10.3390/molecules28196819

**Published:** 2023-09-27

**Authors:** Rong Guo, Jiangnan Yu, Zhikun Guo

**Affiliations:** 1Computational Biology, Bioinformatics and Genomics Program, Department of Biological Sciences, University of Maryland, College Park, MD 20742, USA; 2International Cancer Center, Shenzhen University Medical School, Shenzhen 518054, China

**Keywords:** CD58, colorectal cancer (CRC), virtual screening

## Abstract

Human cell surface receptor CD58, also known as lymphocyte function-associated antigen 3 (LFA-3), plays a critical role in the early stages of immune response through interacting with CD2. Recent research identified CD58 as a surface marker of colorectal cancer (CRC), which can upregulate the Wnt pathway and promote self-renewal of colorectal tumor-initiating cells (CT-ICs) by degradation of Dickkopf 3. In addition, it was also shown that knockdown of CD58 significantly impaired tumor growth. In this study, we developed a structure-based virtual screening pipeline using Autodock Vina and binding analysis and identified a group of small molecular compounds having the potential to bind with CD58. Five of them significantly inhibited the growth of the SW620 cell line in the following in vitro studies. Their proposed binding models were further verified by molecular dynamics (MD) simulations, and some pharmaceutically relevant chemical and physical properties were predicted. The hits described in this work may be considered interesting leads or structures for the development of new and more efficient CD58 inhibitors.

## 1. Introduction

As one of the most common gastrointestinal cancers, colorectal cancer (CRC) ranked as the fourth incidence of malignant tumors and the fifth death rate in China [1]. CRC incidence and death rates have been stabilizing or decreasing in some developed countries, but rapid growth has been seen in many developing countries including China [2].

Increasing evidence suggested that tumor-initiating cells (T-ICs) exist in different tumors, such as brain, breast, prostate, pancreatic, and colorectal tumors [3,4,5,6]. T-ICs are tumor cells that have the ability to regrow the tumor from isolation [7], and are characterized by the distinctive features of self-renewal, proliferation, multi-lineage differentiation, and strong tumorigenicity [8]. Recent studies have revealed that colorectal tumor-initiating cells (CT-ICs) play a crucial role in tumorigenesis, metastasis, recurrence, and treatment resistance of colorectal cancer [9]. The most important and characteristic feature of T-ICs is their increased self-renewal potential [10], which is dominantly regulated by the Wnt pathway in CRC [11]. CD58, a glycosylated adhesion molecule, was identified as a surface marker of CT-ICs [12]. CD58 activation upregulated the Wnt/β-catenin pathway by degradation of Dkk-3 and facilitated the self-renewal ability of CT-ICs. In addition, knockdown of CD58 significantly impaired sphere formation and prevented tumor growth [12].

CD58, also known as lymphocyte function-associated antigen (LFA-3), is a cell adhesion molecule expressed on Antigen Presenting-Cells (APCs) [13]. Protein-protein interaction between CD58 and CD2 can increase the sensitivity of immune recognition, and facilitate the adhesion between T cells and APCs as well as the contacts between cytolytic T cells, natural killer (NK) cells, and their target cells [14,15,16]. It has been shown that CD58/CD2 interaction stimulated the synergistic secretion of CXC chemokine ligand 8 (CXCL-8/IL-8) of human intestinal CD3 + TCRαβ + CD8+ intraepithelial lymphocytes (IELs) [17], and CXCL-8 induces cell proliferation and migration, promoting tumor cell growth in CRC [18,19]. CD58-CD2 interaction is related to maintaining the self-renewal ability of CT-ICs, by promoting the secretion of CXCL-8 from T cells [12]. Therefore, effective blocking of CD58 and intervention of the Wnt pathway were proposed as a potential strategy to treat CRC caused by CT-ICs.

As shown in Figure 1, the extracellular region (171 residues) consists of two immunoglobulin-like domains [20], and domain I is responsible for the adhesion to CD2 [21,22,23]. The crystal structure of a CD2-binding chimeric form of CD58 (Figure 1A, PDB ID: 1 CCZ) reveals that the CD2-binding domain (domain I) has the Ig superfamily V-set AGFCC′C″:DEB domain topology (Figure 1B) and shares several unique structural features with CD2 [24]. Recent findings suggested that domain I is a promising drug target, and some peptides were designed to modulate immune response, targeting the structural epitope of domain I [25,26,27,28,29,30,31,32,33].

Structural-based virtual screening is a computer-aided screening approach to discover novel inhibitors against the selected target by evaluating the chemical structure and binding affinity [34]. High throughput virtual screening is a process that computationally investigates a large set of chemical compounds or materials and discriminates drug candidates from non-candidates by their binding affinities with the target protein [35]. It has become a popular tool for molecular discovery due to the exponential growth of available computational resources and the constant improvement of simulation and machine-learning techniques [36]. Molecular dynamics (MD) simulations can provide important dynamic and structural information about ligand-protein interactions in a flexible manner and has been combined with molecular docking in drug design [37]. In many cases, MD simulations were applied to improve and reinforce the performance of molecular docking [38,39,40].

The study focused on identifying a group of small molecular compounds that have the potential to become CD58 inhibitors from natural products. We screened over 183,000 small molecular compounds from the ZINC database [41,42] and Traditional Chinese Medicine (TCM) database@Taiwan [43]. The screening process was followed by cell assays to test the activities of candidates in vitro. The proposed binding models were further validated by MD simulations, and then ADMET prediction was made to determine the toxicity associated with structural conformation. Five small molecular compounds were identified as potential inhibitors of CD58, and future work would be directed at experimental verification (e.g., cell adhesion inhibition assay to verify the binding models with CD58) and structure optimization of the good candidates.

## 2. Results and Discussion

### 2.1. Active Site Analysis

The extracellular region of CD58 is shown in Figure 1A. Domain I is the CD2-binding domain and the ligand binding sites are located on the AGFCC′C″ interface (Figure 1B). The highly acidic AGFCC′C″ β-sheet interface, containing ten negatively charged residues and six positively charged residues, shows overall electrostatic complementary with the ligands [24]. In addition, as reported, in CD58-CD2′s ‘hand-shake’ binding model, CD58 and CD2 adhesion domains contact each other from opposite ends in an orthogonal orientation, and residues Glu25, Lys29, Lys32, Asp33, Lys34, Glu37, Glu39, and Phe46 are important in the binding [30,44,45] (Figure 2A). Several peptide inhibitors targeting the AGFCC′C″ interface of CD58 have been designed successfully to inhibit cell adhesion and modulate immune response [25,26,27,28,29,30,31,32,33].This provides us with a rational approach to modulate the activation of CD58 and inhibit CD2-CD58 interaction by designing compounds binding to the AGFCC′C″ interface of CD58. Therefore, we selected the AGFCC′C″ interface for virtual screening (Figure 2B), to search for small molecular inhibitors of CD58. The inhibitors could have the potential to suppress the self-renewal potential of CT-ICs and prevent CRC tumor growth by blocking CD58-CD2 interaction as well as the Wnt pathway.

### 2.2. Virtual Screening and Proliferation Inhibition of SW620 Cell Line

A custom high throughput virtual screening pipeline is shown in Figure 3. In the process, both binding affinity from Autodock Vina and commercial availability (e.g., price) were considered to select 9 candidates from NP and 4 candidates from TCM for further cell experiments. To investigate the cytotoxic effects of these compounds on the SW620 cell line (human colorectal adenocarcinoma epithelial cells), cells were incubated with various concentrations of each compound for 48 h, and a CCK-8 assay was applied to analyze cell viability. After the exposure, significant decreases in cell viability were observed following an increased concentration of the compounds. Compared to the approved anti-cancer drug, Nimustine Hydrochloride, five of them (DY6, DY7, DY10, DY11, and DY12) showed relatively good performance, inhibiting the growth of SW620 cells with IC50 values ranging from 26.33 ± 0.23 μM to 104.99 ± 7.86 μM (Table 1, Appendix A).

With resource and funding limitations, only cell assay was performed on SW620 cell lines to validate the inhibition in vitro. However, cell is a system too complex to prove our in silico hypothesis, whether the compounds could selectively bind with the active site of CD58. Future work would be focused on additional experiments to verify the binding models of the candidate molecules. For example, the ability of the candidates to inhibit CD2-CD58 interaction can be evaluated by cell adhesion assay using model systems [31]. In addition, a cell assay on human normal colorectal cells will also be made to test the toxicity.

### 2.3. Binding Analysis of the Five Good Candidates

The possible binding models of the five good candidates (DY6, DY7, DY10, DY11, and DY12) were revealed in Figure 4B–F. As shown in Figure 4A, four of the five hits, including DY6, DY10, DY11, and DY12, fit into the GFCC’ cavity, while DY7 (represented as yellow sticks) interacted with the CC′C″ region. DY7 formed two hydrogen bonds with residues Lys29 and Asp33 of the active site of CD58. The benzene ring interacted with residue Phe46 through a π-π interaction, and the whole skeleton interacted with the key residues Lys29, Asp33, Glu37, Glu39, and Phe46 of the active site through Van der Waals (VDW) interactions.

While the predicted binding models of DY6, DY10, DY11, and DY12 were different from DY7, as they interacted with the GFCC’ region via VDW interactions (Figure 4C–F). All of them formed hydrogen bonds with residues Val26, Glu37, Leu38, Glu39 and Glu78, among which residues Glu37 and Glu39 are key residues [30]. Except for the above interactions, DY10 formed an additional hydrogen bond with residue Glu76. In general, the hydrogen bonds with residues Val26, Glu37, Leu38, and Glu39 stabilized the benzenediol group, and those with residues Glu76 and Glu78 stabilized the saccharide ring. These hydrogen bonds made DY10 stable and well fit into the GFCC’ cavity, forming strong VDW interactions (Figure 4D). The cyclic amide of DY6 formed two hydrogen bonds with the carboxyl of Glu78, which is in the center of the GFCC’ cavity, so the hydrogen bonds may be crucial in the interaction. Moreover, the hydroxyl of the steroid part of DY6 formed a hydrogen bond with the carbonyl of residue Thr83 (Figure 4C). In DY12, the hydroxyl of the saccharide ring formed hydrogen bonds with the carbonyls of residues Ile82, Thr83, and Asp84 (Figure 4E). The only difference between DY10 and DY11 was the orientation of the hydroxyl group in the saccharide ring. As shown in Figure 4D, the saccharide ring of DY10 showed a better electrostatic complementarity with the GFCC’ cavity, which is an electronegative pocket. The negatively charged hydroxyl group of the saccharide ring interacted with the electropositive sides of the pocket in DY10 (Figure 4D), while in DY11, the hydroxyl group formed a hydrogen bond with residue Glu78 and pointed to the electronegative pocket (Figure 4F).

Overall, our binding analysis showed that the five good candidates interacted with the CC′C″ region or the GFCC’ cavity to bind with CD58, and thus could have the potential to affect CD58-CD2 binding and modulate downstream signaling pathways. As three (DY10, DY11, and DY12) of them fit into the GFCC’ cavity and shared the same skeleton, and the only difference was the saccharide ring, we speculate that: the hydrogen bonds with residues Val26, Glu37, Leu38, Glu39, Glu76, and Glu78 are necessary for compound-CD58 binding; the orientation of the hydroxyl groups in the saccharide ring may influence binding affinity through hydrogen bond or electrostatic interactions.

### 2.4. MD Simulations

MD simulation was applied to further investigate the stability of the proposed binding models of the ligand-protein complexes using GROMACS version 2023 [46] at various time points up to 100 ns. Overall, all the complexes were stable throughout the simulation (RMSD range: 0.2–0.5 nm, Figure 5A). Among them, DY6-, DY10- and DY11-CD58 complexes had relatively lower RMSD and RMSF values, which could be explained by their proposed binding models. RMSF analysis provides important information about the flexibility of different regions of the complexes, and the results suggested that all the complexes experienced nearly the same fluctuation throughout the time scale (Figure 5B). ROG spectrum answers a question about the compactness of the system during MD simulation, e.g., the performance of the complex in the biological system [47]. As shown in Figure 5C, the system was seen to sample nearly the same compactness throughout. In addition, we used H-bond functionality in GROMACS to determine whether the hydrogen bonds mediating the ligand-protein interactions in the molecular docking were sustained in the MD simulation. The results indicated that usually 2–4 hydrogen bonds could be generated in the complexes, and as a support of molecular docking and the predicted binding models, DY7 mediated fewer hydrogen bonds than other compounds (Figure 5D).

### 2.5. ADMET Prediction

As shown in Table 2, we estimated various pharmaceutically relevant properties and physical descriptors for ADME profiles for the five good candidates. All compounds showed good ADME parameters. An in silico toxicity risk was also performed to check hERG inhibition, AMES toxicity, carcinogens, and acute oral toxicity. The compounds were found to be of good safety, except compound DY6 was predicted as positive for AMES toxicity. Future work of structure optimization could focus on increasing their activities, as well as decreasing their toxicities.

## 3. Experimental Methods

### 3.1. Library Download

The structural information was collected from the ZINC database [41,42] and Traditional Chinese Medicine (TCM) database@Taiwan [43]. ZINC is a curated collection of more than 230 million commercially available chemical compounds prepared for virtual screening [41,42]. TCM database@Taiwan is a large and comprehensive small molecular database on traditional Chinese medicine for virtual screening [43]. The format of compound structure (mol2 or sdf) was converted to the pdbqt format through Applied Chemistry Software Open Babel 2.3 [48]. To provide potential active compounds for the following structure-based virtual screening, we successfully constructed two 3D structural libraries: traditional Chinese medicines (TCM) library containing 33,765 small molecular compounds from 8445 active ingredients of traditional Chinese medicines; natural product (NP) library containing 149,515 small molecular compounds from the ZINC database. The small molecular compounds were categorized based on their physical and chemical properties (e.g., molecular weight, hydrogen bond donor/acceptor, the number of rotatable bonds, cLogP, etc.), and stored in parallel nodes, to accelerate the computing speed of virtual screening.

### 3.2. Protein Preparation and High Throughput Virtual Screening

The 3D structure of the CD2-binding domain of CD58 [24] (PDB ID: 1 CCZ) was downloaded from the Protein Data Bank (PDB; http://www.rcsb.org/pdb, accessed on 3 August 2015). All water and solvent molecules were removed from the structure, and hydrogen atoms and Gasteiger charges were added using AutoDock Tools [49]. The protein file was prepared in pdbqt format. The grid-enclosing box was set to center on the AGFCC′C″ interface, which is the binding site of CD58 with its ligand CD2. Center coordinates (center_x: −13.24, center_y: 54.06, center_z: 28.1) and box size (size_x: 26, size_y: 22, size_ z: 24) of the grid were proposed to enclose the whole interface. NP and TCM libraries were screened using Autodock Vina [49]. The top 200 compounds (binding affinity range: −10 kcal/mol to −7 kcal/mol) from each library were selected, and the binding models with the residues of the AGFCC′C″ interface were manually analyzed to exclude false positive results. Here, are two criteria for small molecular compounds to be considered as good candidates: shape or molecular electrostatic potential matching with the active site; or having hydrogen bonds with the critical residues (Glu25, Lys29, Lys32, Asp33, Lys34, Glu37, Glu39, and Phe46) of the active site. Their commercial availability and prices were also considered in the screening process. Finally, nine candidates from the NP library and four from the TCM library were chosen for the following cell assay.

### 3.3. Chemistry

All compounds were purchased from two commercial suppliers, Yuanye, Shanghai, China (www.shyuanye.com, accessed on 15 October 2015) and Topscience, Shanghai, China (www.tsbiochem.com, accessed on 15 October 2015), without further purification.

### 3.4. In Vitro Assay on SW620 Cell Lines

Cell viability was measured by the Cell Counting Kit-8 (CCK-8) assay as previously described [50,51]. Briefly, SW620 cells were plated into 96-well plates with 200 μL DMEM medium containing 10% fetal bovine serum (FBS) and 1% Penicillin-Streptomycin solution (PS) at a density of 5 × 10^4^ cells/well. A compound solution was made at the concentration of 2 μL by dissolving with DMEM medium. The cells were incubated with the selected 13 compounds, as well as the positive control Nimustine Hydrochloride [52] (an approved nitrosourea-derived anticancer agent effectively against CRC), at different concentrations (1 mg/mL, 200 μg/mL, 40 μg/mL, 8 μg/mL, 160 ng/mL and 32 ng/mL) in a humidified incubator with 5% CO_2_, 37 ℃ for 48 h. At the end of each treatment, the supernatant was discarded and 200 μL DMEM medium with 10% FBS and 1% PS was added. 20 μL Cell Counting Kit-8 (CCK-8) was added to each well and cells were further cultured for 4 h. The absorbance was measured at a wavelength of 450 nm with a Synergy 2 multimode microplate reader (BioTek, Winooski, VT, USA). The inhibition rate (%) was calculated by the formula:Inhibition% = (1 − F450,compound/F450,control) × 100%

Three biological replicates were made. IC50 values were calculated from the inhibition curves, and the standard deviation was calculated.

### 3.5. MD Simulations

Molecular dynamics (MD) simulation was applied to further validate the dynamics and binding models of the target protein with the five good candidates using GROningen MAchine for Chemical Simulations (GROMACS) version 2023 [46]. CHARMM36 (Chemistry at Harvard Macromolecular Mechanics) was applied as an all-atom force field [53] and CHARMM General Force Field (CGenFF) server was used to retrieve the topology of the hits [54,55]. The complex was put into a 10 nm box. After solvation (TIP3 P water model), neutralization (Na^+^ and Cl^−^ ions), equilibration [canonical (NVT) and isobaric-isothermic (NPT) ensemble for 500 ps], energy minimization was carried out for the neutralized complexes using the steepest descent minimization algorithm and then progressively balanced to 310 K and 1 Bar. For each complex, a 100-ns MD (0.002 ps/step, 50,000,000 steps) was performed on the equilibrated systems.

### 3.6. ADMET Prediction

We evaluated the ADMET properties of the five good candidates using admetSAR, a comprehensive source and free tool for evaluating chemical ADMET properties [56]. It has been considered widely as a useful tool for in silico screening ADMET profiles of drug candidates and environmental chemicals [57,58]. Thirteen properties related to absorption, distribution, metabolism, elimination, and toxicity were estimated for the selected compounds.

## 4. Conclusions

The study screened a group of small molecular compounds as potential CD58 inhibitors from NP and TCM libraries using high throughput virtual screening and binding analysis, and five of the 13 commercially available candidates showed significant proliferation inhibition on SW620 cell lines. MD simulations further verified the predicted CD58 interactions. In silico prediction of ADMET properties indicated that the selected compounds have good pharmaceutical properties and unsatisfactory toxicities. Further experiments (e.g., cell adhesion inhibition assay, cell assay on human normal colorectal cells, etc.) are needed to verify the predicted binding models, and structure optimization could be made to improve their activities and decrease their toxicities. The hits discovered in this work could provide novel scaffolds for further hit-to-lead optimization and lay a foundation for the development of therapeutic candidates for CRC treatments.

## Figures and Tables

**Figure 1 molecules-28-06819-f001:**
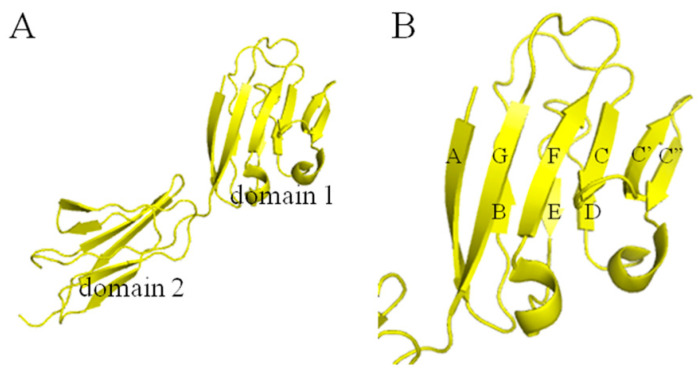
(**A**) The crystal structure of CD58 (PDB ID: 1CCZ). (**B**) Domain I of CD58.

**Figure 2 molecules-28-06819-f002:**
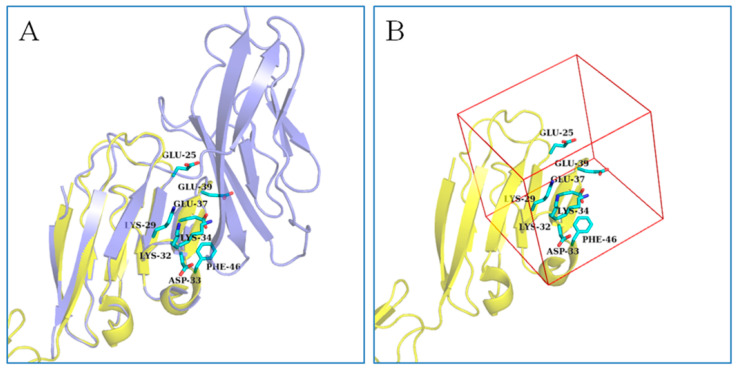
(**A**) The overlay of the CD2-binding domain (domain I) of CD58 (PDB ID: 1CCZ; colored in yellow) and CD58-CD2 complex (PDB ID: 1QA9; colored in purple). Critical residues of the AGFCC′C″ interface were shown as cyan sticks and labeled. (**B**) The selected active site of virtual screening was indicated in the red box.

**Figure 3 molecules-28-06819-f003:**
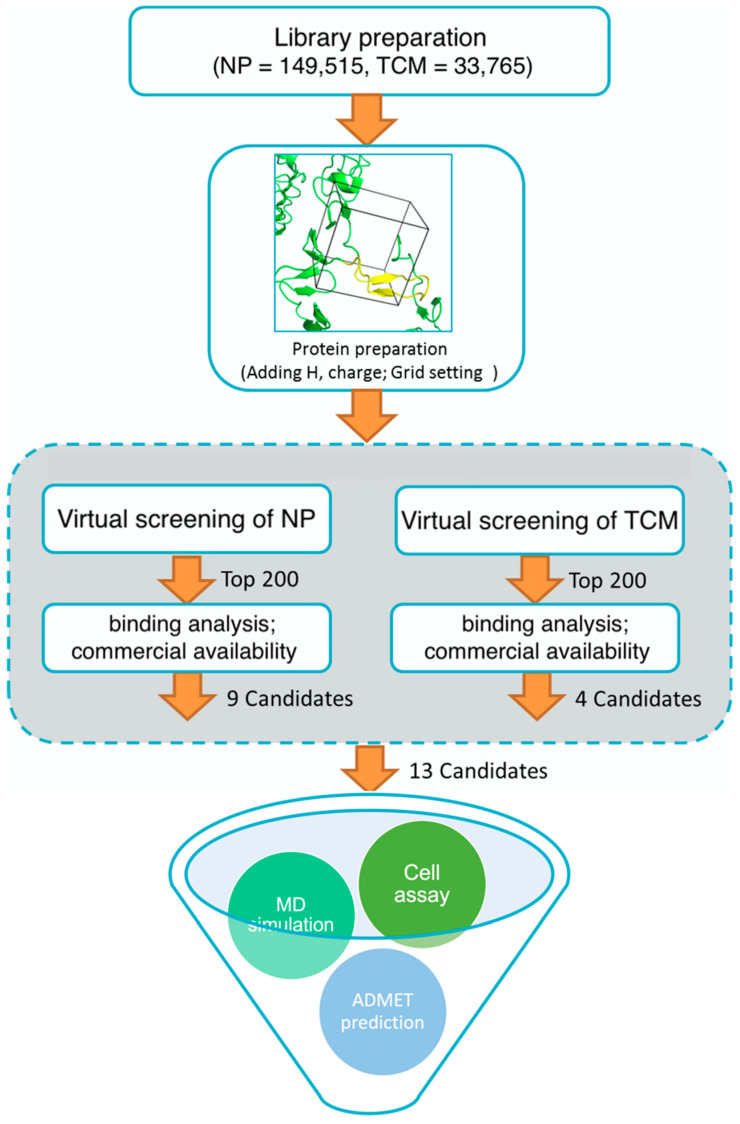
High throughput virtual screening pipeline.

**Figure 4 molecules-28-06819-f004:**
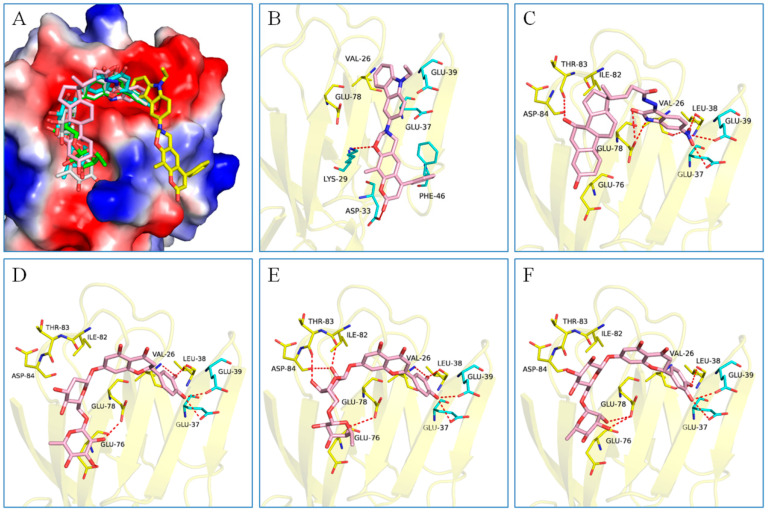
(**A**) Electrostatic potential diagram of CD58 (PDB ID: 1CCZ) interacting with the five good candidates (DY6: pink; DY7: yellow; DY10: white; DY11: cyan; DY12: green). Predicted binding models of DY7 (**B**), DY6 (**C**), DY10 (**D**), DY12 (**E**), and DY11 (**F**) with the GFCC’ cavity of CD58. CD58 was shown as a yellow cartoon, relevant residues were shown as yellow sticks, and the critical residues in binding were colored in cyan. The five compounds were shown as pink sticks. Red dashed lines represented the potential hydrogen bonds.

**Figure 5 molecules-28-06819-f005:**
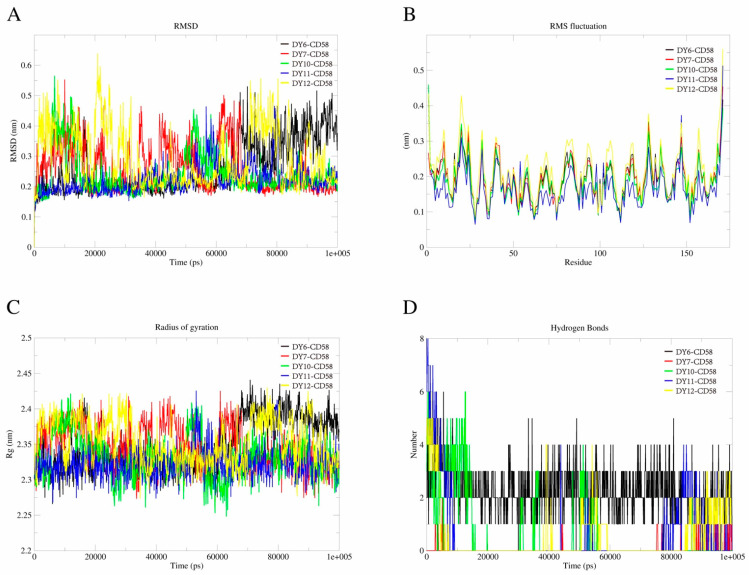
MD simulation results of the DY6-CD58, DY7-CD58, DY10-CD58, DY11-CD58, and DY12-CD58 complexes in 100 ns. (**A**). The RMSD (Root Mean Square Deviation) plot. (**B**). The time-evolving RMSF (Root Mean Square Fluctuation) plot. (**C**). ROG (Radius of gyration) spectrum. (**D**). Hydrogen bond distribution.

**Table 1 molecules-28-06819-t001:** Structures and IC_50_ values of the 13 candidates.

Compound	MW ^a^	Affinity (kcal/mol)	Structure	SW620 Cell Line ^b^
				IC_50_ (μM)
DY1	496.5	−9.0	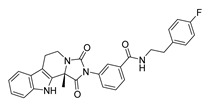	>200
DY2	575.7	−9.2	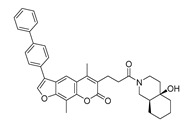	>200
DY3	412.4	−8.8	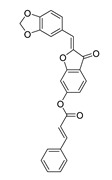	>200
DY4	482.5	−8.7	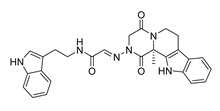	>200
DY5	630.7	−8.8	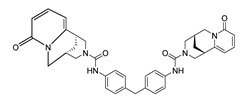	>200
DY6	580.7	−9.0	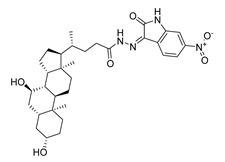	31.30 ± 0.88
DY7	486.6	−8.9	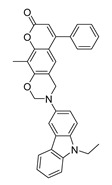	89.38 ± 15.28
DY8	511.6	−9.0	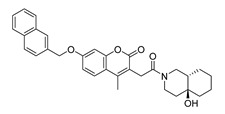	>200
DY9	722.0	−8.8	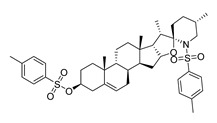	>200
DY10	580.5	−8.3	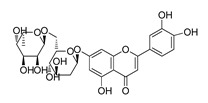	26.33 ± 0.23
DY11	596.5	−8.2	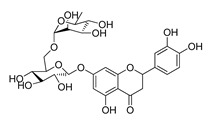	104.99 ± 7.86
DY12	594.6	−8.5	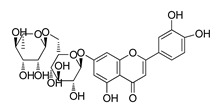	59.44 ± 3.40
DY13	594.5	−7.6	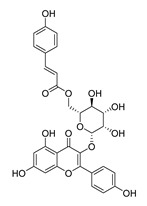	>200
Nimustine Hydrochloride	309.2		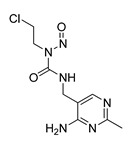	59.44 ± 4.45

^a^ Molecular weight. ^b^ SW620 cell line is a human colorectal adenocarcinoma epithelial cell line.

**Table 2 molecules-28-06819-t002:** In silico prediction of ADMET properties.

ADMET Properties	DY6	DY7	DY10	DY11	DY12
BBB	BBB−	BBB+	BBB−	BBB−	BBB−
HIA	HIA+	HIA+	HIA+	HIA+	HIA+
Pgp Substrate	Substrate	Non-substrate	Substrate	Substrate	Substrate
hERG Inhibition	Non-inhibitor	Non-inhibitor	Non-inhibitor	Non-inhibitor	Non-inhibitor
AMES Toxicity	AMES toxic	Non AMES toxic	Non AMES toxic	Non AMES toxic	Non AMES toxic
Carcinogens	Non-carcinogens	Non-carcinogens	Non-carcinogens	Non-carcinogens	Non-carcinogens
Acute Oral Toxicity	low toxic	low toxic	low toxic	low toxic	low toxic
Molecular Weight	580.73	486.57	580.54	596.54	594.52
LogP	5.25	7.25	−0.43	−1.46	−1.39
Rotatable Bonds	6	3	6	6	6
H Bond Acceptors	10	5	14	15	15
H Bond Donors	4	0	8	9	9
Surface Area	161	213.67	233.03	237.82	236.11

## Data Availability

Data is contained within the article or Appendix A.

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
