# Peer review of "Virtual Screening and Binding Analysis of Potential CD58 Inhibitors in Colorectal Cancer (CRC)"

_molecules, 2023, doi:10.3390/molecules28196819_

Round 1
Reviewer 1 Report (Previous Reviewer 2)
Despite I think that the binding assays are crucial for this kind of work, I appreciate the insertion of MD studies to support the docking hypothesis.
However, I suggest the acceptance of the manuscript after minor changes:
-The title should be changed in "Virtual Screening and Binding Analysis of POTENTIAL CD58 Inhibitors in Colorectal Cancer (CRC)" as the CD58 activity was not tested.
-In the MD paragraph bibliographic data supporting the utility of MD simulation in reinforcing docking analysis must be introduced (e.g. DOI: 10.3390/molecules26041103, https://doi.org/10.3389/fchem.2021.661230).
Author Response
Please see the attachment.

Reviewer 2 Report (Previous Reviewer 1)
The manuscript by Guo et al. described the discovery of CD58 inhibitors by high throughput virtual screening. 13 candidates were selected and followed up with cellular toxicity study, ADMET prediction, and MD simulation. Some of the candidates showed good pharmaceutical properties. The following points need to be addressed:
1. Section 2.3 Chemistry: Include the HPLC traces of the leading hits in supplementary information.
2. The authors tested the leading candidates’ toxicity effect in human colorectal adenocarcinoma epithelial cells. However, the cellular toxicity might be due to off-targets effect. Further studies are needed to show the target engagement in cells or with pure protein.
Author Response
Please see the attachment.

Reviewer 3 Report (New Reviewer)
In the manuscript entitled, " Virtual Screening and Binding Analysis of CD58 Inhibitors in Colorectal Cancer (CRC)", authors have obtatined a ligand library and screened it computationally against CD58 receptor for identifying the potential anticancer agents followed by their ADMET profiling, MD simulation and cell line based in vitro assay. Overall, the research presents valuable insights into the development of CD58 antagonists and their potential role in alleviating colorectal cancer. Here are some suggestions to improve the quality of the manuscript:
1. Docking protocol needs to be validated with the appropriate ligand to enhance the authenticity of the computational prediction.
2. Section 2.1 Library Preparation, here preparation term is inappropriate, it should be changed with download or procured
3. The use of SW620 cell line should be justified with respect to the concerned receptor CD58 which is missing in the manuscript.
4. The quality of the figures should be improved for clarity of the obtained results.
5. The comparative result analysis for the in vitro experimental outcome with their computational analysis and justified for their deviations.
6. Conclusion should be rephrased by giving more emphasis of the obtained results
Incorporating these review comments should help enhance the overall quality and impact of your research article. Good luck with your revisions and the submission process.
Minor editing of English language required
Round 2
Reviewer 2 Report (Previous Reviewer 1)
Thank you for addressing the comments. It's necessary to include all HPLC traces to show the purity (over 90%) especially you put all the compounds in cells.
Author Response
Please see the attachment.

This manuscript is a resubmission of an earlier submission. The following is a list of the peer review reports and author responses from that submission.
Round 1
Reviewer 1 Report
The manuscript by Guo et al. described the discovery of novel CD58 inhibitors by virtual screening. Over 183,000 compounds were screened against the CD-2 binding domain of CD58 and 13 leading molecules were further tested with a cell viability assay in human colorectal adenocarcinoma epithelial cells. ADMET properties of the leading molecules were predicted by admetSAR. The following points need to be addressed:
1. As the authors mentioned in the manuscript “cell is a system too complex to prove our in silico hypothesis, whether the compounds could selectively bind with the active site of CD58.”, the viability experiment alone cannot show the binding of the candidates to CD58. Additional experiments are needed to show the binding, such as lymphocyte-epithelial assay and E-rosetting assay.
2. The authors mentioned that “enzymatic assays to verify the binding models of the candidate molecules. ” Be more specific about the assays.
3. Page 4, section 3.1: In “Gokhale, Ameya, et al. "Immunosuppression by Co-stimulatory Molecules: Inhibition of CD 2-CD 48/CD 58 Interaction by Peptides from CD 2 to Suppress Progression of Collagen-induced Arthritis in Mice." Chemical biology & drug design 82.1 (2013): 106-118.”, Glu25 was not considered as an important residue for the binding. More references are needed for this claim.
4. Page 2: “Recent findings suggested that domain I is a promising drug target, and some peptides were designed to modulate immune response, targeting the structural epitope of domain I.” There are also some other newly reported peptides targeting CD58 to be referred here.
5. Figures and figure captions can be improved. For example, Figure 2A is the overlay of the structure of CD2 binding domain and CD58-CD2 complex. Show the distances of the potential hydrogen bonds in Figure 4.
6. Add the HPLC traces of the leading hits in SI.
7. Formatting issue in reference.
Minor points:
1. Discuss the Figure 1 in main text.
2. “While the predicted binding models of DY6, DY10, DY11 and DY12 were different from DY7, as they interacted with the GFCC' region via VDW interactions (Figure 4B-4F).” It should be 4C-4F.
Minor editing of English language required
Reviewer 2 Report
The manuscript of Guo et al. describes the application of a structure-based virtual screening strategy to identify new CD58 inhibitors for the cure of colorectal cancer. Despite the scope of the study is of interest for the anticancer research field, the outcomes do not fully support the aim of the paper. Indeed, the authors were able to only test the anti-proliferative activity of the compounds selected from the virtual screening which is not enough to validate the in silico hypothesis as the same authors assessed in the manuscript.
Major revisions:
The CD58 inhibitory activity should be provided.
Minor revisions:
-In the experimental section, details about the preparation of the 3D databases used for the VS should be provided.
-How many poses were generated for each ligand during the virtual screening with Autodock Vina?
The quality of english language is good.
Round 2
Reviewer 1 Report
Thanks for addressing the comments. Again, the anti-proliferative activity is not sufficient to validate the hypothesis. Binding activity needs to be tested.
Reviewer 2 Report
I appreciate the efforts made by the authors, but binding assays are crucial to validate in silico hypothesis.
English language is good